# Microfluidics-Based Systems in Diagnosis of Alzheimer’s Disease and Biomimetic Modeling

**DOI:** 10.3390/mi11090787

**Published:** 2020-08-19

**Authors:** Yan Li, Danni Li, Pei Zhao, Krishnaswamy Nandakumar, Liqiu Wang, Youqiang Song

**Affiliations:** 1Energy Institute, Qilu University of Technology (Shandong Academy of Sciences), Jinan 250014, China; liyan@sderi.cn (Y.L.); zhaop@sderi.cn (P.Z.); nandakumar@lsu.edu (K.N.); 2School of Energy and Power Engineering, Qilu University of Technology (Shandong Academy of Sciences), Jinan 250014, China; 3Department of Neurology, Jinan Central Hospital, Cheeloo College of Medicine, Shandong University, Jinan 250013, China; ywc85695211@163.com; 4Cain Department of Chemical Engineering, Louisiana State University, Baton Rouge, LA 70803, USA; 5Department of Mechanical Engineering, Faculty of Engineering, The University of Hong Kong, Hong Kong, China; 6School of Biomedical Sciences, The University of Hong Kong, Hong Kong, China; 7State Key Laboratory for Cognitive and Brain Sciences, The University of Hong Kong, Hong Kong, China

**Keywords:** Alzheimer’s disease (AD), microfluidic chips, AD biomarkers, blood–brain barrier (BBB), three-dimensional AD model

## Abstract

Early detection and accurate diagnosis of Alzheimer’s disease (AD) is essential for patient care and disease treatment. Microfluidic technology is emerging as an economical and versatile platform in disease detection and diagnosis. It can be conveniently integrated with nanotechnology and/or biological models for biomedical functional and pre-clinical treatment study. These strengths make it advantageous in disease biomarker detection and functional analysis against a wide range of biological backgrounds. This review highlights the recent developments and trends of microfluidic applications in AD research. The first part looks at the principles and methods for AD diagnostic biomarker detection and profiling. The second part discusses how microfluidic chips, especially organ-on-a-chip platforms, could be used as an independent approach and/or integrated with other technologies in AD biomimetic functional analysis.

## 1. Introduction

### 1.1. Alzheimer’s Disease (AD) Neuropathology and Significance for Early Diagnosis and Disease Modelling

Alzheimer’s disease is one kind of chronic neurodegenerative disease. The dementia develops slowly and gets worse over time. Once diagnosed, damage of the brain and loss of body function are irreversible [1]. Early detection and diagnosis of AD is very important for disease management. However, current procedures in AD diagnosis are tedious and require mutual corroboration [2]. There is great necessity to develop rapid, accurate, and non-invasive methods for AD diagnosis. On the other hand, traditional AD neuropathological research used to employ either cell culture models or animal models in elucidating neurodegenerative mechanisms [3,4]. Such studies have been providing valuable in vivo scenarios in AD analysis. But researchers need to pay special attention in handling animal models which is labor-intensive and time-consuming, and it is difficult to conduct real-time monitoring and precise control of the biological processes in animal experiments. Furthermore, there might be discrepancies between humans and animal models due to the species difference. To address these limitations, researchers are developing micro-engineered organ-on-a-chips that could recapitulate the architectures of complex biological tissue for application in AD physio-pathological studies.

### 1.2. Advantages of Microfluidic-Chip-Based Systems in AD Research

Microfluidic-chip-based systems have been used extensively in small volume sample preparation, separation, mixing, purification, detection, and assay [5]. It could be designed and manufactured according to the natural structure and function of tissues and organs. Micro-engineered organ-on-a-chips have been developed as a powerful and efficient tool in biological and pathological research by integrating lab-on-a-chip (LOC) devices [6,7]. With miniatured sizes that are comparable with human tissues, and chambers for compartmentation and co-cultured cells in a 2D or 3D manner [8,9], LOC might reproduce the physical architectures and microenvironments of human tissue. The chip materials are mostly biocompatible polymers (such as polydimethylsiloxane (PDMS) and gelatin) and are transparent thus enabling high-resolution and real-time imaging in various biological scenarios. Therefore, microfluidic-chip-based systems could provide a functional tissue and organ context for in vitro living cell and biomedical analysis. All these make it a suitable platform in AD diagnostics, neuropathological studies, and new drug developments [10,11,12,13].

In this review, we first focus on the recent application of microfluidic technology in AD biomarker detection and analysis; then, a brief overview of how microfluidic organ-on-a-chip could be used as a powerful platform in AD disease physio-pathological study, especially in blood–brain barrier (BBB) modelling, and therapeutic drug screening is provided.

## 2. Biomarkers Detection in AD Pathology Study

Biomarkers are biological indicators of the body’s physiological status which could reflect the biological processes under normal/pathological conditions and/or pharmacologic responses to drug treatment. The ideal biomarkers should be sensitive and specific, clinically reliable, reproducible, simple to perform, inexpensive, and non-invasive. In AD diagnosis, traditional diagnostic procedure (for example enzyme linked immunosorbent assay (ELISA)) is tedious, time-consuming, and could not detect AD in the early stages. Therefore, as a good alternative to ELISA, a biomarker-based approach plays pivotal roles in AD diagnosis, especially in preclinical AD detection [14,15,16,17].

In the last decades, there was considerable growth in the investigations on candidate biomarkers in cerebrospinal fluid and plasma. It is now clear that Aβ (especially Aβ42), tau, and phospho-tau are widely accepted cerebrospinal fluid (CSF) biomarkers, and they could facilitate clinical diagnosis (including early diagnosis) and aid in gaining important information in AD neuropathological processes. Other emerging new biomarkers, such as neurofilament light chain and non-coding miRNAs, also attracted research interests in their usefulness in aiding disease progression prediction.

### 2.1. Histopathological Biomarkers Detection

Amyloid-beta protein (especially Aβ_42_) and Tau protein (and phosphorylated tau protein) are the most characteristic AD histopathological biomarkers; they showed good correlation with AD pathology [14]. They are the most studied biomarkers in AD detection and diagnosis [15,18] (Figure 1).

#### 2.1.1. Aβ Characterization and Profiling

Abnormal Aβ generation and deposition in the hippocampus region is one of the characteristic pathological hallmarks of AD. AβAmyloid β-protein is most concentrated in the CSF and is also found in other body fluids such as saliva, blood, nasal mucosa, and the lacrimal gland. As it is comparatively harder to obtain CSF samples, examination of other body fluid Aβ levels would be a valuable alternative for disease status identification. Different sample sources (e.g., saliva, blood urine, or CSF) may contain different Aβ peptides format (e.g., Aβ_40_ or Aβ_42_, oligos or polymers) [19,20,21]. It is also important to identify in advance which Aβ peptides format in the sample is mostly associated with disease.

Typical measurement of Aβ is done using ELISA technique, which is labor-intensive and time-consuming. Emerging studies that applied a microfluidic technique in Aβ characterization and profiling (also relying on the principle of antigen-antibody reactions) have found improved sensitivity, reduced time, and sample amount in the measurements. Mohamadi et al. (2010) [22] described a microfluidic capillary electrophoresis (MCE) chip for rapid characterizing of CSF Aβ peptides. They modified the inner surface of the PDMS channel to control electroosmotic flow (EOF) and to reduce surface adsorption. Separation of five synthetic Aβ peptides and determination of the relative abundance of Aβ_1-42_ were studied using this microchip. The system was also validated with CSF samples from AD and non-AD patients; two Aβ peptides (Aβ_1-40_ and Aβ_1-42_) could be detected but not quantified. Their study presented a facile method to study the native state of Aβ peptides (both Aβ_1-40_ and Aβ_1-42_) and found that the oligomerization status was also important in disease diagnosis. They further developed the system for more sensitive capture and detection of Aβ peptides [23]. This time a nano-porous membrane was integrated for preconcentration of the samples. The MCE channel was revised, and Western blotting was replaced with fluorescent detection which was more sensitive and faster. They proved that their system worked well for analyzing truncated Aβ peptides which is much more important for early detection of AD (Figure 1A).

Another group also reported a PDMS microfluidic device for on-chip determination of amyloid polypeptide–Aβ_42_ for point-of-care testing in early AD diagnosis [24]. Their device was integrated with a miniatured quartz crystal microbalance (mQCM) resonator and Aβ_42_ antibody was immobilized on the mQCM surface by cross-linking. On-chip determination of Aβ_42_ was realized by measuring the frequency response during the antigen–antibody binding process. The linear range of detection was from 0.1 μM to 3.2 μM for serum Aβ_42_. The detection limit was as low as 0.1 μM, a value that was close to the AD patient’s cut-off value and important for early AD diagnosis. The device was also tested on specificity and reusability, which all proved this device could be a reliable point-of-care testing clinical application.

Based on immunoassay and droplet microfluidics, a single-step magnetic-beads-based droplet-wise approach was developed for molecular diagnosis of AD [25]. The sandwich assay complex included magnetic beads grafted with capture antibodies, the detection antibodies and the monomeric Aβ peptides and were used for CSF sample examination. To detect trace Aβ peptides in the sample, a sequential microfluidic droplet setup was constructed which included sample droplets, magnetic bead droplets, washing droplets, and, finally, detection droplets. The forward movement of droplets was realized by pairs of magnetic tweezers. Through this single sequence, Aβ peptide concentrations with smaller sample volumes (>1 μL) could be successfully detected and higher throughput could be achieved (Figure 1B).

#### 2.1.2. Tau Protein (and Phosphorylated Tau Protein) Measurement

Tau protein and phosphorylated tau (pTau) protein were associated with another AD pathological hallmark, known as neurofibrillary tangles (NFTs). Phosphorylation of tau protein triggers NFT formation, and the formation of NFTs destroys neuronal skeleton stabilization [26]. Destabilization of the neuronal skeleton might contribute to neuronal cell break down which finally leads to neuronal degeneration [27]. The CSF levels of tau were positively associated with the degree of AD dementia [28]. Clinical evaluation of tau and pTau also depended on the ELISA kit [29]; there was great need for developing more sensitive and faster measurement technique.

Based on immuno-reaction, Vestergaard et al. (2008) [30] demonstrated a localized surface plasmon resonance (LSPR) method and for on-chip detection of tau protein. They first fabricated a multi-spot LSPR nano-particle chip and immobilized tau-mAb on the chip. Tau protein was then introduced onto the chip and non-specific adsorption was blocked by Tween-20. The changes in the thickness of the gold-capped silica nanoparticle layer surface could be reflected in absorbance increase. Then, an optical signal could be detected and recorded by a UV-Vis spectrometer. The detection limit was as low as 10 pg/mL, lower than the CSF tau protein cut-off value. They also proved this immune chip was highly specific for tau protein (Figure 1C).

#### 2.1.3. Neurofilament Light Chain as a New Blood-Based Biomarker

Neurofilament light chain (NfL) is a sub-unit of neurofilament, and the dominant component of axonal cytoskeleton [31]. Studies of animal models and multiple neurological disorders have provided strong evidence that body fluidic NfL changes contribute to brain damage and brain atrophy [31,32,33]. With the help of a single molecular array platform (Quanterix), serum NfL was precisely evaluated and directly linked to pre-symptomatic Alzheimer’s disease and proved to be a useful blood biomarker for disease progression and brain neurodegeneration prediction [34].

### 2.2. Genetic Markers Detection

Amyloid precursor protein (*APP*), presenilin 1 (*PS-1*), presenilin 2 (*PS-2*), and apolipoprotein E (*APOE*) are widely accepted genetic risk factors for AD [35]. Amyloid precursor protein, *PS-1*, *PS-2* mutations are identified by sequencing method; therefore, we did not discuss them here. We only focused on *APOE* analysis, as it was the most common and important genetic modifier.

Different from histopathological biomarkers, genetic markers usually depended on the ELISA method and required CSF samples. The fluorescence in situ hybridization (FISH)-based technique was an effective system in detecting genetic factors for AD, especially when using blood, saliva, and urine samples [36]. Recently, the combination of microfluidic system with the FISH technique proved to be a more efficient, sensitive, and faster method in disease detection and diagnosis [37,38,39]. Yang et al. (2014) [40] reported a “microE-DASH” chip for allele-specific genotyping of the *APOE* gene. They used a fluorescence-based solution-phase method called dynamic allele-specific hybridization (DASH) which can accurately discriminate between homozygous and heterozygous SNP genotypes to perform electrochemical melt-curve measurements. By monitoring real-time thermal melting curves of the duplexes formed by surface-bound probes and their DNA targets, their device could achieve multiplexed detection of homozygous and heterozygous genotypes at multiple loci in a single step. The principle and method for SNP detection might be available for other SNP analyses and integrated with microfluidic PCR and single-strand generation for SNP-based disease diagnostics.

Besides genotypic analysis, apolipoprotein E (APOE) in human plasma could also be detected at protein level by microchips. Mariana (2014) [41] took advantage of on-chip immunoassays and used electrochemical reactions of quantum dots (QDs) as labels for sensitive and efficient detection of APOE. In this study, a microfluidic platform for Alzheimer’s biomarker detection was set-up using quantum dots as electrochemical labels. The performance of the microfluidic platform was first evaluated by a model protein and then applied in APOE sample detection. Using this method, they found the limit of detection for APOE concentrations was 12.5 ng/mL, and the linear detection range was 10–200 ng/mL. The APOE concentration results were comparable with other measurements, demonstrating the feasibility of this approach as a possible point-of-care device.

### 2.3. MicroRNA as Biomarkers in AD

While considerable advancements have been made in developing proteinic and genetic biomarkers for AD diagnosis, other small molecular biomarkers, such as the non-coding RNA (ncRNA), were attracting worldwide research interest [42]. MicroRNAs (miRNAs) are a family of 19–24 nucleotide ncRNA and function as important posttranscriptional gene regulators [43]. Studies in cancer have resolved that miRNAs can function as reliable biomarkers [44] which suggested possibilities for using miRNAs in other pathological conditions including AD [45].

Accumulating data have shown that circulating miRNAs were promising biomarkers to monitor AD pathogenesis [46,47,48,49,50,51], and researchers are employing novel analytical platforms in miRNAs biomarker discovery [47,52]. However, there were discrepancies among current studies for reasons including restricted sample sizes, different disease progression stages, sample sources variations (e.g., blood, serum, plasma, CSF, saliva, urine), different human populations, un-uniform methodological processes, etc. Comprehensive studies with large data sets (involving different human populations) and conducted under standard methodological procedures might be able to better illustrate the role that circulatory miRNAs play as a peripheral biomarker.

## 3. Microfluidic Platform as a New Approach in AD Physio-Pathological Analysis

Besides AD biomarkers characterization and quantification, microfluidic chips are also powerful tools in unraveling AD physio-pathological processes (Figure 2). The advantages of using microfluidic platforms in AD studies include: (1) providing physical compartmentation with chip size comparable to that of brain; (2) being compatible with cell culture and thus creating oxygen and nutrition gradients for chemotaxis analysis; (3) applying dynamic flow stresses; and (4) most chips being optical and thus enabling real-time and on-chip visualization [46]. Researchers can now investigate the detailed processes of Aβ transmission [53], aggregation [54,55], neurotoxicity [56,57,58], clearance [59], interactions between Aβ and microglia [60], and also tau pathology [61,62] as summarized in Table 1.

### 3.1. Microfluidic Models for AD Physio-Pathological Study

#### 3.1.1. Amyloid beta Pathology

##### Amyloid Beta Transmission in Neurons

Pathological evidence showed that as AD progresses, Aβ may be propagated to distal regions through neuroanatomical connections [66]. Unravelling the transmission mechanism may help to inhibit disease progression. To study the pathway of Aβ transmission in neurons, Song et al. (2014) [53] applied PDMS microfluidic chip as culture chambers and investigated how Aβ pathology was transmissible through neuronal networks. The chambers contained compartments for culturing different cells which were connected by microchannels. Axons could extend to the compartment through these microchannels. The authors successfully showed that Aβ was transmitted via axonal processes, and retrogradely transported to neuronal cell bodies and then to adjacent neurons. And the result was also confirmed in mouse brain from in vivo β-amyloid injection and by Li et al. (2017) [58]. Their study would serve as useful suggestions for novel therapeutic strategies to block or slow down Aβ transmission and disease progression.

##### Amyloid Beta Aggregation

Learning the mechanism of Aβ peptide assembly is critical for understanding the biophysics of amyloidosis [67]. Researchers utilized the small volume and confined space of microchannels and developed a microfluidic method to analysis environmental factors on beta-amyloid aggregation [54]. Compared to a bulk system, a microfluidic system could generate uniform fibrils with shorter incubation time, which could also be used for screening of Aβ aggregation inhibitors. Further, Meier et al. [55] modified the surface of a microfluidic device in order to construct a miniatured system to control Aβ aggregation. Their report was valuable in that they overcame the interfacial effects on miniaturization of aggregation experiments for small sample volumes.

##### Amyloid Beta Aggregates Clearance

Since abnormal deposition and aggregation of Aβ is the major neuropathological event in AD, dissociation and clearance of Aβ aggregates is one of the approaches to treat and prevent AD. Lee and Park (2010) [59] developed an efficient microfluidic system for Fe^3+^-induced Aβ aggregates clearance using metal chelators. They compared the clearance outcomes of different metal chelators in the microchannels and found deferoxamine was the most effective one. Deferoxamine might play its role through destabilizing the fibrillar structures of Aβ aggregates which disassembled into Aβ monomers or oligomers and flowed out of the microchannel. Such destabilization was caused by Aβ aggregates secondary structure changes. Their study offered an efficient pre-screening method for chemical candidates which could enhance the clearance of Aβ deposits before in vivo analysis.

##### Amyloid Beta Neurotoxicity

Several studies of Aβ fibrillogenesis on neurotoxicity have suggested that it was soluble Aβ oligomeric assemblies rather than fibrillar Aβ proteins which possess neurotoxicity [68,69,70,71]. Choi et al. (2013) [56] constructed a microfluidic platform that could provide a brain bio-mimetic condition and generate continuous flow/gradient of oligomeric assemblies of Aβ and analyzed the on-chip neurotoxicity of Aβ oligomeric assemblies. The results did not support that an increase in the number of fibrils had significant neurotoxic effects on neurons and that the neurotoxicity of oligomeric assemblies might be more pronounced than fibrils. This was confirmed by Ruiz et al. (2014) [57], in which they constructed an open culture microchip for the evaluation of Aβ’s effect on primary central nervous system (CNS) cells. They tested the toxicity of both oligomeric and fibrillar Aβ and found oligomeric Aβ was more toxic than the fibrillar form. To further illustrate how Aβ peptide affects neuronal function within different subcellular structures, Li Wei et al. (2017) [58] utilized a microfluidic compartmentalized device with isolated chambers for separate cultures of axons, dendrites, and cell body and analyzed the subcellular localization and toxicity of Aβ peptides. Their device could generate gradients of chemotactic factors along the grooves that guided the outgrowth of axons and dendrites toward opposite directions. Through this model, they found that Aβ_1–42_ oligomer was much more neurotoxic on axons than on dendrites, and that Aβ_1–42_ could induce calcium dynamics in neurons. They also found that Aβ_1–42_ was transported along axons from the distal terminals and transmitted to the cell bodies in a bidirectional manner. Besides neurotoxicity analysis, these microfluidic platforms could also function as useful tools for neuron cell culture and toxicity and drug efficacy assessment.

#### 3.1.2. Microglial Activation

Microglias are critical nervous system-specific cells which play major roles in brain development, maintenance of neural environment, response to injury, and repair. In the context of AD, microglia would accumulate in the vicinity of Aβ plaques and then became activated [72]. It is important to study how and why microglias migrate and were stimulated by Aβ for therapeutic purpose. Cho et al. (2013) [60] constructed a microfluidic chemotaxis platform system to mimic how and why microglia accumulates in the vicinity of Aβ plaques. Their microfluidic chip was annular and axisymmetric with surface-bound Aβ inside the central compartment (CC) and surrounded by migration channels (MC) and the annular compartment (AC). Soluble Aβ of monomers and oligomers were filled between the CC and concentration gradients were created. Then dosage dependent microglia migration was monitored by recording fluorescence image of microglial accumulation inside the model. Through this model, the Aβ effective concentration and inhibition concentration for microglial migration could be determined. As a model system, this platform was also applicable to studies of other cell types’ migration where the potency of cytokines and chemokines need to be quantified.

#### 3.1.3. Tau Pathology

To fully capture tau-related pathological mechanisms for AD modifying therapeutics, it is important to understand the processes of tau internalization/uptake, transfer, and propagation. Microfluidic technology could provide a unique platform than traditional culture dish for neuronal cell culturing and analysis since the micro-chambers could realize compartmentalization and micro-grooves could facilitate the outgrowth of axons or dendrites [46]. Taylor et al. (2003, 2005) [63,64] reported a neuronal culture micro-device with two compartments separated by a physical barrier. Cells were seeded in the soma chamber and micro-grooves in the barrier allowed outgrowth of neurites. This and other work paved the way for on-chip-culturing of neuronal cells and allowed for facile analysis of tau uptake and transport in axons and/or dendrites [46,61,64]. Microfluidic devices could also be used to study tau transfer/propagation when cells were synaptically connected [65]. Another study revealed that tau propagation could be enhanced by increased neuronal activity [62]. Microfluidic platform had enabled new investigations of tau pathology in AD as well as high-resolution live imaging of neuronal cells.

### 3.2. Blood–Brain Barrier (BBB)

The BBB protects the central nervous system (CNS) by providing essential compartmentation to prevent toxic substances from entering the brain, while still maintains selective permeability to regulate substance transport from the blood to the brain and vice versa [73].

The key features of the BBB include: (1) Brain microvasculature are mainly composed of three cell types, cerebral endothelial cells, astrocyte end-feet, and pericytes (PCs); (2) The endothelial cells link by tight junction proteins form the primary structure and surrounded by astrocytes, pericytes; (3) As one microvasculature unit, they function together and are responsible for BBB selective permeability and maintenance of high transendothelial electrical resistance (TEER); (4) Endothelial cells are sensitive to shear stress from fluid flow and the mechanotransductive effects would influence cell differentiation and tight junction formation [74,75].

Due to the selective permeability of BBB, most therapeutic drugs cannot cross the barrier. Since it is unethical to have tests on human, it is essential to develop both in vivo and in vitro BBB models not only for therapeutic drug screening and evaluation, but also for physio-pathological analysis of the brain.

For an ideal BBB model, structural similarity, biological analogous and functional feasibility are the most desired features. Classical in vivo BBB models are mostly constructed with laboratory animals, which could provide natural physiological and pathological environment and having been providing valuable information for scientific research. However, it is usually time-consuming and labor-intensive to develop animal models, while there is always discrepancy due to species difference between humans and other animals. Developing in vitro BBB models and using human-derived brain cells would be promising alternatives compared to in vivo ones. And the advantages of in vitro BBB models are direct, simple and easy to control.

The early in vitro cellular BBB models were constructed with static Transwell chambers [76,77], which were comparatively straight forward and easy to handle. Other studies were using dynamic BBB models [78,79,80,81,82], which could provide information not only on shear force induced endothelial cell changes but also on tight junction intactness [83]. With the rapid development of micro-technologies, researchers have been exploring the potentials to construct microfluidic BBB models to be used as disease models and/or drug screening platform [84,85], as summarized in Figure 2.

Booth and Kim (2012) [79] developed a multi-layered microfluidic blood brain barrier (μBBB) device that comprised a sandwiched polycarbonate membrane, four PDMS components, and two glass out layers (Figure 2B). The assembled PDMS device formed two perpendicular flow channels to mimic the dynamic environment. Endothelial cells and astrocytes were cultured, respectively, on the lumenal and ablumenal sides of the semi-porous membrane (10 μm) to form a dual-layer BBB. Morphological and functional measurements could prove their model was an effective in vitro model system for studies of barrier function and drug delivery.

Griep et al. (2013) [86] constructed an improved BBB chip (Figure 2C) based on the work of Douville et al. (2010) [87]. They used human brain endothelial cells in forming monolayer. The cells could be cultured up to 7 days in the chip and generated trans-epithelial-electric-resistance (TEER) value that was comparable to reported static models, and also tight junction protein expression level. To study the effect of shear stress on endothelial cell physiology, the cells were first cultured under static condition for 3 days and then exposed to flow induced shear stress. This caused a three-fold increase of the TEER value which proved its function as a mechanical barrier. The authors also showed that biochemical modulation of the chip model correlated with BBB dysfunction, further confirming it was a dynamic BBB model.

To provide a real-time image of both the vascular and neuronal sides of the BBB, Prabhakarpandian et al. (2013) [88] engineered a synthetic microvasculature model with dimension and microenvironment comparable to that of in vivo BBB (Figure 2D). Their chip was composed of side-by-side apical and basolateral chambers separated by pillars for communication between the two chambers. Diverging and converging microchannels were designed to mimic the bifurcation and junction features of microvasculature. Endothelial cells were cultured in the apical chamber, and astrocytes were kept in the basolateral chamber to provide astrocyte-conditioned medium. Real-time permeation studies and transporter assay all showed their BBB model could function as a selective permeability barrier.

To include all human cell types, Brown et al. (2015) [89] designed an in vitro BBB model called Neuro Vascular Unit (NVU) (Figure 2E). The NVU device involved neurons, astrocytes, pericytes and endothelial cells in the fabrication, which formed a vascular chamber, a brain chamber and separated by a porous membrane. Independent perfusion of both compartments was separated by the membrane. This design allowed for cell-to-cell communication between chambers and separate delivery of drugs and nutrients to either chamber, which formed a facile platform for function modeling of the BBB as well as for drug toxicity and permeability test. Other researchers used induced pluri- and multi-potent stem cells in modeling co-cultured NVU also achieved structural and functional similarity and proved suitability for drug development [90].

The above-mentioned models were usually sandwich chips with living cells lined on the two sides of a membrane or such surrogate, to better recapitulate the 3D constitution of the BBB in vivo and provide a multiple physiological microenvironment [91], Wang et al. (2016) [92] constructed a 3D model by co-culturing endothelial cells, pericytes, and astrocytes in a layered microfluidic chip (Figure 2F). The brain endothelial cells and pericytes were cultured on opposite sides of a porous membrane, which was a bi-culture model and mimicked the natural in vivo organization. The astrocytes were cultured on the bottom of the lower channel to construct the triculture model. Their model showed high viability of all cultured cells up to 21 days. The TEER and permeability assays all showed comparable results to natural BBB. Functional expression of the P-glycoprotein efflux pump was also correlated with the increase in the number of days in culture. These all provided robust testimony that their tri-culture model is functional in vitro BBB model.

Three-dimensional (3D) printing technique offers an alternative approach for BBB modeling. Kim et al. (2015) [93] created an array of engineered 3D brain microvascular structure in the middle of a collagen I matrix by utilizing 3D printing (Figure 2G). Mouse brain endothelial cells (bEnd.3) were cultured on the luminal surface of cylindrical collagen microchannels for reconstruction of brain microvasculature in vitro with circular cross-sections. They studied the barrier function of brain microvasculature by transendothelial permeability test and disruption/recovery and proved their 3D microvascular a useful model for both physiological and pathological testing.

To give an example of how in vitro BBB model could be applied in neuropathological analysis and therapeutic drug screening, Xu et al. (2016) [94] advanced the BBB model by introducing a dynamic 3D PDMS-based microfluidic system (Figure 2H). Their microfluidic chip model was arrayed with independent functional units connected by micro-channels. Each functional region consists of one vascular compartment for introducing fluidic flow and one brain compartment with channels for infusing natural extracellular matrix (ECM) collagen or brain cells. Primary rat brain cells were then introduced and co-cultured with pericytes and astrocytes to form a continuous cell layer. Barrier integrity and function and impermeability tests all proved that this model could emulate the structural, functional and mechanical properties of in vivo BBB. This model could be readily applied to studies that require cell-cell, cell-matrix and cell-signaling factors interactions.

To further address how BBB dysfunction was associated with neuropathological processes, Shin et al. (2019) [95] developed a 3D microfluidic platform to investigate cerebral–vascular interactions in AD. Their model could reproduce several key aspects during AD pathogenesis, including increased BBB permeability, decreased tight junction protein expression, abnormal Aβ deposition as well as increased inflammation and oxidative reactions. Therefore, it could be useful as a physiological relevant model for BBB impairment analysis and a well-controlled drug-screening platform.

Besides being the neuronal barrier, high selectivity for substance transport is another important physiological function of BBB. Most recently, there were significant progresses in analyses of BBB transport function. Park et al. (2019) [96] used hypoxia-induced brain microvascular endothelial cells (BMVECs) to develop a BBB model with physiological relevant permeability and transporter properties. This enhanced BBB model expressed multiple efflux pumps and transporter proteins and could regulate the traffic of therapeutic drugs and antibodies across the BBB. To better illustrate drug transport and distribution in the vascular and perivascular regions, Ahn et al. [97] engineered a BBB chip with brain endothelial cells, pericytes, and 3D astrocytic network. They demonstrated their BBB model showed key BBB features including low permeability, decreased reactive astrogliosis and polarized expression of AQP4, which is critical to maintain water and ion homeostasis in the brain. Further, they showed on-chip mapping of nanoparticle transport and distribution at both cellular and tissue levels, which paved the way for quantitative evaluation of therapeutic drug distributions as well as targeting efficacy.

### 3.3. 3D Co-Culture Models

Since 3D cell culture models could offer many advantages over two-dimensional (2D) models, researchers were exploiting the possibility of micro-engineered system in analyzing all aspects of AD physiological and pathological processes as well as therapeutic drug screening. Bianco et al. (2012) [98] reported an overflow microfluidic network for studying neuronal cell-cell interactions under various stress conditions. By seeding different brain cells in separate microfluidic chambers and independently controlling culturing conditions and selectively priming single cell types with inflammatory stimuli, they were able to successfully dissect brain-region-specific glial contribution in different in vitro models of neuro-inflammation and unravel the underlying molecular mechanisms involved in the cross-talk among different cell populations.

Lee and colleagues [99,100] did a series of advanced work on neuronal cell 3D culture modelling and application in a neuro-system study [99,100]. Choi et al. (2013) [99] developed a PDMS model based on concave microwell array. The concave molds were size controllable and neurospheroids were formed in the microconcave wells. Immuno-staining confirmed that the neuronal cells formed neural network among neurospheres. This 3D model was applied to test the neurotoxicity of amyloid beta and the results showed it could serve as a promising tool for toxicity testing and/or drug screening. Park et al. (2015) [100] took one step forward to mimic brain in vivo conditions by creating a 3D cyto-architecture and interstitial flow model. They found that compared to static culture model dynamic flow model could enhance the growth/formation of neural network and differentiation of neural progenitor cells into neurons. They also tested neurotoxic effects of amyloid-β on neurospheroids and showed potential of the brain-on-a-chip model as a drug-screening and cytotoxicity testing tool. Given the versatility of these in vitro micro-models, they could readily be applied to other neuro-associated disease studies, like Parkinson’s disease, brain tumor, infection, and stroke.

### 3.4. Prospect Models for AD Study

Despite all the effort for in vitro AD model, one fact that could not be ignored was brain endothelial cells might lose brain-specific properties under in vitro culture environment over time, thus limiting the reproducibility of the BBB models. There was good news that under low-adherence conditions, co-culture of human brain ECs, pericytes, and astrocytes could spontaneously form into a multicellular spheroid and self-assemble into a BBB-like modular construction [101]. Cho et al. (2017) [102] modified the culturing method and formed a highly structured multicellular spheroids for recapitulation of BBB key barrier functions, including the expression of tight junctions, molecular transporters and efflux pumps. This has been evolved into “BBB organoids” for a versatile and cost-effective platform for therapeutic drug discovery [103]. By using induced pluripotent stem cell (iPSC), it is expected that “brain organoids” can readily differentiate themselves into different brain cell types, such as neurons, microglia, and glia cells, to establish organoids to model a “brain tissue” (Figure 3) for individual AD patients pathological analysis as well as “individualized” drug development [104].

## 4. Conclusions and Future Perspectives

In this review, we briefly summarized the latest developments and applications of microtechnology in AD diagnosis and biomimetic modelling. Under the circumstances, Alzheimer’s disease ranks first among the most commonly observed neurodegenerative disorders; early detection and accurate diagnosis of AD still prove to be a formidable task ahead of us. At the same time, comprehensive understanding of the etiology of AD would lead to better preventive measurement and therapeutic outcome.

In order to successfully develop AD biomarker detection approaches, a variety of microfluidic devices were modeled for characteristic histopathological markers analysis with reduced testing time/sample amount, enhanced specificity and reproducibility. Despite these advantages, researchers might need to put more efforts in developing parallel and high-throughput techniques for large-scale sample screening. The current results need to be tested in larger data set and compared with standard measurement to reach a reliable conclusion. Another limitation is that microfluidic methods were mainly effective for dissociated molecules. For analysis of associated membrane proteins (such as PS-1, PS-2), it might need to be integrated with other measurements to produce the best possible results.

On the other hand, microfluidic chip, especially organ-on-a-chip, provided an alternative choice for disease pathological modeling and therapeutic drugs testing. The microfluidic platform could provide a miniatured model with size, structure, and cell culture comparable to that of the brain, and a basic physiological microenvironment with oxygen/nutrition gradient and dynamic flow stress. Coupled with reliable and sensitive detection measurement, it could enable real-time and on-chip examination for disease pathological processes and response to external stimuli. Nowadays, researches on Aβ transmission/aggregation/clearance/neurotoxicity, tau pathology, neuronal cell-cell interactions and BBB malfunction and related pathological processes could all be realized through microfluidic system. The next step would be the involvement of patient-derived cells in microfluidic models together with improved biochemical and mechanical setups for both genotype and phenotype specific studies. Being an inter-disciplinary and fast-involving area, microfluidic-based systems are expected to fundamentally change the way from basic science to translational drug discovery in AD neurodegenerative studies.

## Figures and Tables

**Figure 1 micromachines-11-00787-f001:**
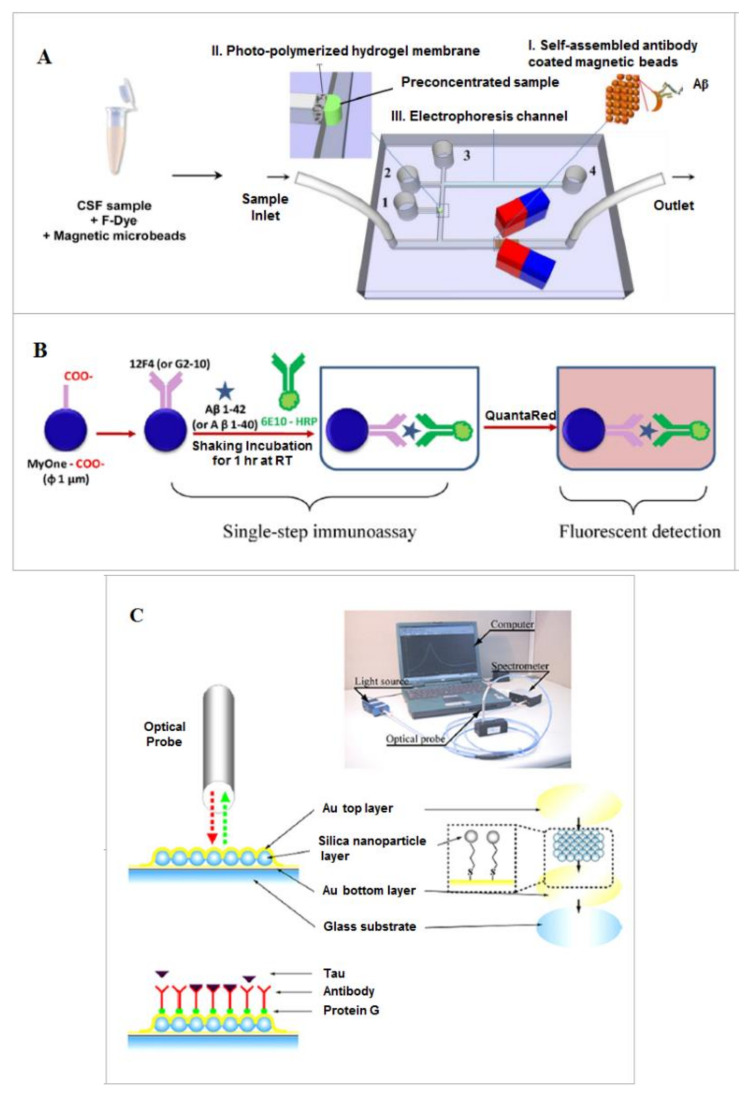
Characteristic Alzheimer’s disease histopathological biomarkers detection using microfluidic technology. (**A**) An integrated microfluidic chip for β-amyloid peptides immunocapture, preconcentration, and separation: (I) plug of self-assembled magnetic beads coated with antibodies inside the microchannel for immunocapture of the target peptide (Aβ), (II) photopolymerized hydrogel plug for preconcentration of the eluted sample, and (III) microchip electrophoresis unit coupled with fluorescence detection (reservoirs 1 to 4 were used for applying voltage during preconcentration and electrophoresis steps). Reproduced with permission from Reference [23], AIP Publishing. (**B**) Magnetic beads-based immunoassays of Aβ1–40 and Aβ1–42. Reproduced with permission [25]. (**C**) Localized surface plasmon resonance (LSPR) for tau immunoassay. Reproduced with permission [30].

**Figure 2 micromachines-11-00787-f002:**
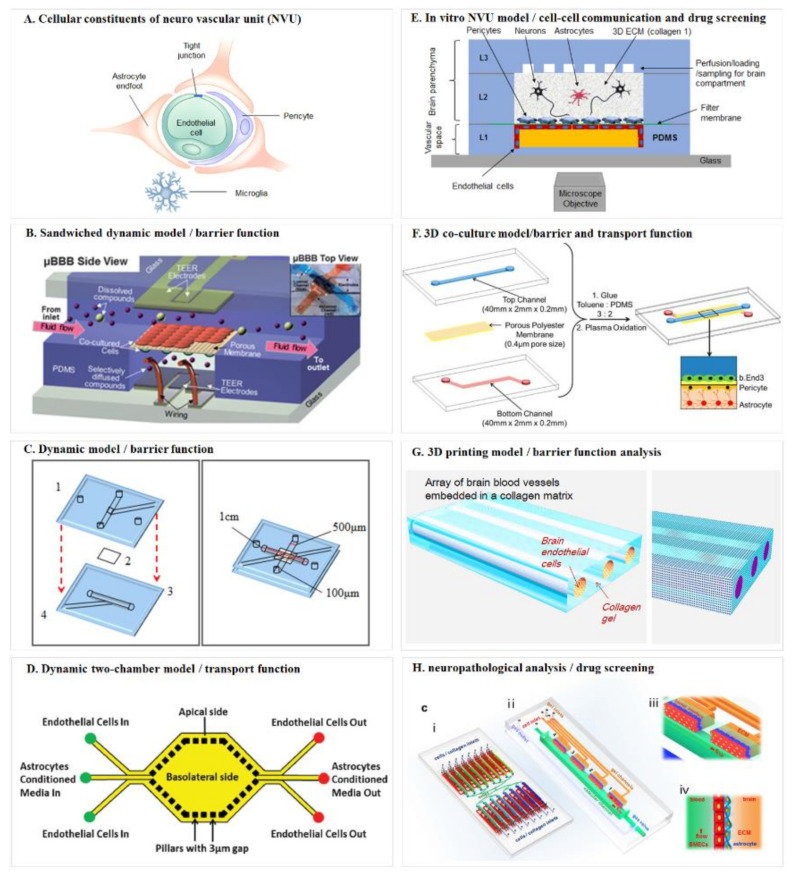
Schematic diagrams of the blood–brain barrier mimicking microfluidic chip. (**A**) Cellular constituents of the neuro vascular unit (NVU). Adapted from Reference [75]; (**B**) sandwiched dynamic model. Reproduced with permission from The Royal Society of Chemistry [79]; (**C**) dynamic model/barrier function. Adapted from Reference [86]; (**D**) dynamic two-chamber model/transport function. Reproduced with permission from The Royal Society of Chemistry [88]; (**E**) in vitro NVU model/cell–cell communication and drug screening. Adapted from Reference [89]; (**F**) 3D co-culture model/barrier and transport function. Reproduced with permission from ACS publications [92]; (**G**) 3D printing model/barrier function analysis. Adapted from Reference [93]; (**H**) Neuropathological analysis/drug screening. Reproduced with permission from Springer Nature [94].

**Figure 3 micromachines-11-00787-f003:**
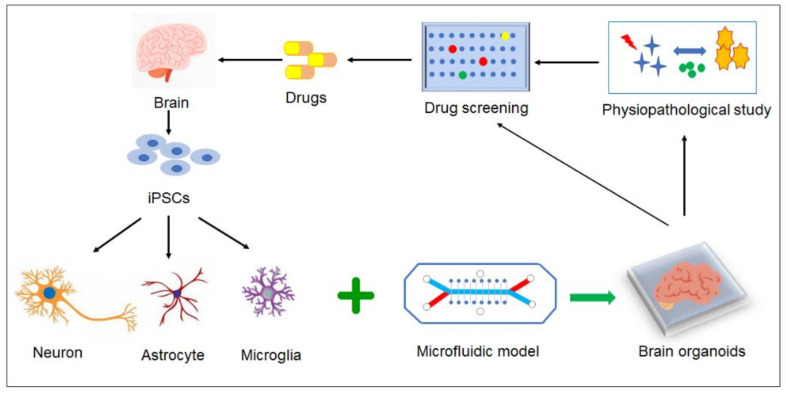
Prospective models for AD study. By using iPSC, “brain organoids” can differentiate themselves into different brain cell types, such as neurons, microglia, and glia cells, to establish organoids to model a “brain tissue” for individual AD patients pathological analysis as well as “individualized” drug development.

**Table 1 micromachines-11-00787-t001:** Microfluidic approaches in AD physio-pathological analysis.

Physio-Pathological Process	Methodology	Cell Type (Culture Time)	References
Aβ transmission	Polydimethylsiloxane (PDMS) microfluidic culture chambers connected by microchannels	Rat cortical neuron (14 days)	[53]
Aβ aggregation	PDMS microchannels Plug-based microfluidics	-	[54,55]
Aβ aggregates clearance	PDMS microchannels	-	[59]
Aβ neurotoxicity	PDMS microfluidic chip PDMS microfluidic chip consisted of cell body and neurite compartments connected by microgrooves	Rat primary neurons (3 days)	[57,58]
Microglial activation	PDMS microfluidic chemotaxis platform	Human microglial cells (9 days)	[60]
Tau pathology	Microfluidic chamber devices with compartmentalization and micro-grooves	Mouse primary neurons; Human induced pluripotent stem cells (over 20 days)	[61,62,63,64,65]

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
