# Peer review of "Microfluidics-Based Systems in Diagnosis of Alzheimer’s Disease and Biomimetic Modeling"

_micromachines, 2020, doi:10.3390/mi11090787_

Round 1

Reviewer 1 Report

  • This review paper is very well reviewed the recent developments and trends of microfluidic applications in AD research. The first part is on the principles and methods for AD diagnostic biomarker detection and profiling. The second part discuss how microfluidic chips, especially organ-on-a-chip platform, could be used as an independent approach and/or integrated with other technologies in AD biomimetic functional analysis. 
  • Unfortunately, I think that tabulating biomarker detection methods would be a great review paper.

Author Response

Many thanks to reviewer 1 ‘s comments and suggestion.

The authors are all agree that AD biomarker detection could readily make one big review paper. AD biomarker detection was the first and important part of this manuscript, we have included histopathological markers, genetic markers, microRNA markers. However, we were only focusing on the basic concept and methodology of microfluidic application in AD biomarker detection, we did not extend to all the details in the manuscript. Also, please kindly be reminded that Table 1 was for ‘Microfluidic approaches in AD physio-pathological analysis’, not for AD biomarker detection.

The English language had been checked by a native speaker and grammar as well as type errors had been corrected.

Reviewer 2 Report

This review highlights recent advances in the use of microfluidic systems to diagnose and model Alzheimer's disease. The authors' focus is primarily on the use of microfluidic-based sensing platforms for detecting biomarkers in biological samples. They also discuss existing organs-on-chip systems that have been developed to understand the pathophysiology of the disease. Overall, the manuscript is timely and has useful information for the diverse readership of Micromachines. However, English is poor. I found numerous grammatical errors and typos as I was reading the manuscript. The quality of the Figures is also inferior. The fonts are tiny and are not readable. Additionally, the authors should add a section to the manuscript to discuss available or potential biomarkers for early diagnosis of Alzheimer's disease. 

Author Response

Many thanks to reviewer 2 ‘s comments and suggestions.  

  1. The English language of this manuscript had been checked by a native speaker and the corrections were made and highlighted in the revised manuscript by using track change.
  2. The authors had revised figures (figure 1, figure 2 and figure 3) in the manuscript to make them clearer to the readers. Original figures were uploaded in the system.
  3. One paragraph discussing available biomarkers for early diagnosis of Alzheimer's disease was added in line 78-84.